# A Robust Approach Assisted by Signal Quality Assessment for Fetal Heart Rate Estimation from Doppler Ultrasound Signal

**DOI:** 10.3390/s23249698

**Published:** 2023-12-08

**Authors:** Xintong Shi, Natsuho Niida, Kohei Yamamoto, Tomoaki Ohtsuki, Yutaka Matsui, Kazunari Owada

**Affiliations:** 1Graduate School of Science and Technology, Keio University, Yokohama 223-8522, Japan; shixintong@ohtsuki.ics.keio.ac.jp (X.S.); niida@ohtsuki.ics.keio.ac.jp (N.N.); 2Department of Information and Computer Science, Keio University, Yokohama 223-8522, Japan; yamamoto@ohtsuki.ics.keio.ac.jp; 3Atom Medical Co., Tokyo 113-0021, Japan; yutaka.matsui@atomed.co.jp (Y.M.); owada.kazunari@gmail.com (K.O.)

**Keywords:** fetal heart rate, Doppler ultrasound, signal quality assessment, autocorrelation function, unsupervised representation learning

## Abstract

Fetal heart rate (FHR) monitoring, typically using Doppler ultrasound (DUS) signals, is an important technique for assessing fetal health. In this work, we develop a robust DUS-based FHR estimation approach complemented by DUS signal quality assessment (SQA) based on unsupervised representation learning in response to the drawbacks of previous DUS-based FHR estimation and DUS SQA methods. We improve the existing FHR estimation algorithm based on the autocorrelation function (ACF), which is the most widely used method for estimating FHR from DUS signals. Short-time Fourier transform (STFT) serves as a signal pre-processing technique that allows the extraction of both temporal and spectral information. In addition, we utilize double ACF calculations, employing the first one to determine an appropriate window size and the second one to estimate the FHR within changing windows. This approach enhances the robustness and adaptability of the algorithm. Furthermore, we tackle the challenge of low-quality signals impacting FHR estimation by introducing a DUS SQA method based on unsupervised representation learning. We employ a variational autoencoder (VAE) to train representations of pre-processed fetal DUS data and aggregate them into a signal quality index (SQI) using a self-organizing map (SOM). By incorporating the SQI and Kalman filter (KF), we refine the estimated FHRs, minimizing errors in the estimation process. Experimental results demonstrate that our proposed approach outperforms conventional methods in terms of accuracy and robustness.

## 1. Introduction

Fetal heart rate (FHR) serves as a vital metric for assessing the well-being of a fetus. It aids in identifying high-risk pregnancies and reducing fetal mortality rates. Cardiotocography (CTG) is a commonly employed method that allows the simultaneous recording of both maternal and fetal heartbeats. Among the techniques used to extract FHR from CTG recordings, the Doppler ultrasound (DUS) technique has gained prominence [1]. This involves attaching an ultrasound (US) transducer to the mother’s abdomen to enable continuous monitoring, as depicted in Figure 1. Previous studies [2,3] have established that the most accurate FHR estimation relies on invasive fetal electrocardiogram (ECG), with non-invasive fetal ECG representing the second most precise method. While FHR estimation based on DUS signals does not attain the same level of accuracy as fetal ECG, it offers distinct advantages. Notably, invasive fetal ECG is restricted to specific cases due to the undesirability of long-term attachment of fetal scalp electrodes for FHR monitoring. Conversely, non-invasive fetal ECG signals obtained by placing electrodes on the mother’s abdomen necessitate expensive equipment and expert operation [4]. To capture a DUS signal, the ultrasound (US) transducer is positioned on the mother’s abdomen. The primary technique for estimating fetal heart rate (FHR) from these DUS signals is based on the autocorrelation function (ACF) [5,6,7]. This method leverages the inherent periodicity of the DUS signals to extract the FHR information accurately. Another notable approach for FHR estimation involves empirical mode decomposition (EMD). By employing EMD, it becomes possible to identify signal components associated with the valve motion of the fetal heart [8,9]. The ACF-based approaches have demonstrated superior consistency and applicability compared to EMD-based methods, primarily due to their simpler computations and fewer parameters. However, current approaches still face several challenges that warrant attention. Firstly, prior studies [5,6,7] have predominantly focused on a single domain (time or frequency) during ACF calculations, potentially resulting in the loss of valuable information from the other domain. Moreover, DUS signals are susceptible to various interferences, including maternal, fetal, and instrument movements, leading to a degradation in the signal-to-noise ratio (SNR). Accurately estimating FHR using noisy DUS data becomes challenging, as the quality of the fetal DUS signals significantly impacts the accuracy of the FHR estimation [10]. To enhance FHR estimation accuracy, a common technique employed is signal quality assessment (SQA), which involves identifying segments of low-quality signals and subsequently eliminating or interpolating erroneous FHR values derived from these segments. Presently, the available literature on signal quality assessment (SQA) methods for DUS signals remains limited, with only a few approaches published [10,11]. However, these approaches based on supervised machine learning techniques usually require large labeled datasets, and there are few public DUS datasets with professional annotations. Furthermore, the existing SQA methods primarily focus on removing the estimated fetal heart rates (FHRs) from the identified segments of low-quality DUS signals. However, this approach may result in a reduced proportion of reserved FHRs in relation to the overall number of estimated FHRs within a record. Hence, alternative strategies are required to address this issue effectively.

Inspired by the shortcomings of existing DUS-based FHR estimation and DUS SQA methods, we present a robust DUS-based FHR estimation method supported by DUS SQA based on unsupervised representation learning. By means of a band-pass filter and a short-time Fourier transform (STFT), the DUS signals are pre-processed in the form of spectrograms. The spectra are integrated over the spectrograms to produce integrated spectral data. The ACF is applied to a 3.75 s integrated spectrum segment and a peak is detected. An approximate fetal RR interval (FRRI) is first determined from the detected ACF peak, which is used to resize the window for the second ACF calculation. The detected peak of the second ACF is defined as the FRRI of that segment. In addition, the length of the window movement is determined based on the FRRI estimated from the previous window. Our DUS SQA uses a fully convolutional network (FCN)-based variational autoencoder (VAE) to obtain learning representations of pre-processed DUS signals by integrating the autoencoder (AE) models mentioned in [12,13]. By feeding these learned representations into a self-organizing map (SOM), we generate a combined signal quality indicator (SQI) that incorporates the quantization error (QE) of the SOM [14]. The combined SQI is used to modify the measurement noise covariance, and a Kalman filter (KF) is then used to correct the inaccurate FRRIs computed from low-quality signals [15,16]. We would like to mention that our DUS SQA method has been previously reported [17], serving as a foundation for our current research. In this paper, we utilize the DUS SQA method proposed in our previous work [17]. However, the fetal heart rate estimation section is entirely novel and presented for the first time. Additionally, we include a comparative analysis of the experimental results between our approach and a conventional method, which remains unpublished.

We conducted a thorough evaluation of the proposed method using DUS recordings obtained from ten subjects. The experimental results clearly demonstrated that our method outperformed the conventional approach [7] in terms of FHR estimation accuracy. Furthermore, the integration of our DUS SQA method led to a substantial reduction in the estimation errors of FHRs. The main contributions of our work can be summarized as follows:We combine time and frequency information using STFT for DUS signal pre-processing so that the ACF-based FHR estimation algorithm does not focus only on a single domain as in previous works.This method is robust to DUS recordings of different qualities because it is supported by DUS SQA.An unsupervised representation learning-based DUS SQA approach is proposed in this paper, which eliminates the need for a large dataset of quality labels. Furthermore, representation learning enables our method to exploit deeper information than human-defined features.

The structure of this paper is as follows. Section 2 summarises and discusses related work. Subsequently, the preliminary work is presented in Section 3. The proposed method is described in Section 4. In Section 5, we evaluate the performance of our proposed method and discuss its strengths and limitations. Finally, we conclude this paper in Section 6.

## 2. Related Work

Some FHR estimation techniques using DUS signals have been reported [5,6,7,8,9]. ACF-based FHR estimators are the most commonly used, which have been shown to be computationally inexpensive and can achieve accurate FHR estimation. A two-step approach for FHR extraction using DUS signals has been proposed by Peter et al. [5], based mainly on the ACF technique. The cardiac cycle times are roughly estimated based on low-pass filtering of the envelope signals, and then the FHRs are calculated using ACF in the frequency domain. However, they mentioned that this method has a relatively high sensitivity to signal-to-noise ratio (SNR), which means that low SNR can lead to inaccurate results. Jezewski et al. [6] presented an ACF-based technique for FHR estimation from DUS signals, which includes three main steps: dynamic adjustment of the ACF window, adaptive ACF peak detection, and determination of beat-to-beat intervals. The ACF windows are continuously adjusted based on the most recent FHR estimate. In addition, if ACF peaks fall within the previously identified FHR range, they are assigned a higher probability of correlating with the correct cardiac cycle. To assess the quality of DUS signals, they also propose that a lower ACF peak indicates lower signal quality, which increases the possibility of inaccurate estimation of cardiac cycle duration. Although Jezewski et al. [6] introduced an indicator to reduce the negative effects of low-quality signals, a single indicator is not reliable enough to remove all low-quality signals. Another research paper [7] provides a generalizable, reproducible, open-source ACF-based technique that can accurately estimate FHR from 1D-DUS obtained using a low-cost handheld transducer. A total of 721 DUS signal segments of 3.75 s length were used to train an FHR estimator, and the parameters of the estimator were adjusted by Bayesian optimization. In addition, it has been mentioned in this paper that only high-quality signal segments are selected for experiments based on the SQA method proposed in [10]. However, by using a rectangular window, the ACF was calculated only in the time domain, so some information in the frequency domain may be lost. Moreover, the fixed window length for ACF calculation limits the accuracy of FHR estimates.

In addition to ACF-based approaches, the empirical mode decomposition (EMD) method has also been used for FHR estimation. Rouvre et al. [8] proposed that the performance of FHR estimation can be improved by applying ACF to the intrinsic mode functions (IMFs) after empirical mode decomposition. However, they also mentioned that this method has not yet been fully formalized theoretically and that the computational complexity is highly dependent on the intrinsic signal properties. Furthermore, an EMD kurtosis method for FHR extraction from DUS signals was presented by Al-Angari et al. [9]. In their work, the kurtosis function is calculated on the FHRs extracted from the DUS signal. It is worth mentioning that the most important indicator of this approach is the appropriate window size used to apply the kurtosis function. It has been shown that the EMD-based methods have better performance than the ACF-based methods when processing signals with relatively high SNR, in other words, the EMS-based methods are more robust. However, it is very possible that some related parameters, including the number of IMFs and the size of the window used to calculate kurtosis, may be overfitted during optimization due to the limited amount of data.

Compared to EMD-based techniques, ACF-based methods show higher reproducibility and practicality due to the relatively low computational complexity and a small number of parameters. However, there are still several drawbacks to the existing methods that can be improved:The ACF calculations in these existing works only focus on one domain, time, or frequency. Thus, the information about the other domain may be lost during the calculation.The performance of ACF-based FHR estimation techniques is highly dependent on the quality of the DUS signal. Although some papers [6,7] use simple SQA approaches to solve this problem, these DUS SQA methods can still be enhanced.

So far, a small number of SQA methods for DUS signals have been introduced. A DUS SQA based on the Support Vector Machine was proposed by C. E. Valderrama et al. [10]. Some innovative template-based SQIs derived from correlation coefficients between DUS signals and the fetal heartbeat-based template signals are used for DUS SQA. The Naive Bayes classifier was also used for DUS SQA [11], where twelve SQIs were used. These SQIs are mostly related to the range of valve movement. Although it has been demonstrated that these DUS SQA approaches can discriminate between different quality levels, there are still certain limitations:Large labeled datasets are usually required for these research works based on supervised machine learning methods. However, there are few DUS datasets with quality annotations. In addition, annotation of DUS quality levels is laborious and requires expert knowledge.The human-defined signal quality features used in these works limit the ability to mine deeper signal quality information in DUS signals.These existing methods simply eliminate the estimated FHRs from the detected low-quality DUS signal segments, which may result in a reduction in the proportion of reserved FHRs to all estimated FHRs in each recording.

Although not for DUS signals, some unsupervised learning-based SQA methods have been introduced to improve signal analyzing performance. A paper [18] evaluates photoplethysmography signal quality by computing seven SQIs associated with entropy and waveform morphology and training a SOM for quality-level classification. In addition, two AE-based ECG SQIs associated with reconstruction errors and reconstruction reliabilities have been proposed by N. Seeuws et al. [12]. Recently, unsupervised representation learning has gained popularity for time series classification tasks. In the paper [19], time series data can be transformed into an instance-feature matrix using a proposed effective unsupervised representation learning methodology, where they also demonstrated that these features can be used to perform precise time series clustering tasks. J. Pereira et al. [13] have also proposed a VAE-based representation learning strategy for anomaly identification. How to appropriately apply the SQA results for achieving better FHR estimation accuracy is another crucial task, in addition to the evaluation of the signal quality. Li et al. [15] introduced a method that utilizes KF to correct heart rate estimates, incorporating four ECG SQIs to adaptively adjust the measurement noise covariance. In addition, for non-invasive fetal ECG signals, another paper [16] also uses KF to enhance the precision of FHR estimation.

## 3. Preliminaries

### 3.1. Doppler Ultrasound (DUS) Signal

For continuous recording, the US transducer is placed on the maternal abdomen to extract DUS signals, as shown in Figure 1. When US waves are reflected from an object, the frequency of the US waves changes, which is called the Doppler frequency shift. This frequency shift, $f_{s}$, is given by:(1)$$f_{s}=\frac{2f_{0}v\cos\theta }{c},$$
where *c* is the US propagation velocity and $f_{0}$ is the transmission frequency. θ is the angle of the object along the direction of the US wave, and vcosθ is the velocity of the object along that direction [4]. The US waves propagate through the mother’s skin and some tissues. The transmitted US waves are reflected by the fetal chest wall. During reflection, the frequency of the US waves changes due to the movement of the heart wall and valves of the fetal heart and, in some cases, blood flow. The reflected US waves are received by the transducer on the mother’s abdomen. The received US waves contain such fetal physical movements, including heart activity, and thus it is possible to estimate the FHR by analyzing the received US wave, i.e., a DUS signal.

### 3.2. Autocorrelation Function (ACF)

It provides a statistical representation of the similarity between a time series and its delayed version. By utilizing the ACF, the analysts can compare the current value of a dataset with its past values. This function operates by comparing a given time series with a lagged version of itself across one or multiple time periods [20]. For a given time series data $\left[Y_{1},Y_{2},\ldots ,Y_{N}\right]$, the ACF with a lag of *k* is defined as follows:(2)$$acf_{k}=\frac{\sum_{i=1}^{N-k}\left(Y_{i}-\bar{Y}\right)\left(Y_{i+k}-\bar{Y}\right)}{\sum_{i=1}^{N}{\left(Y_{i}-\bar{Y}\right)}^{2}}.$$

### 3.3. Variational Autoencoder (VAE)

Autoencoder (AE), which is a neural network consisting of an encoder and a decoder, can extract representations from the inputs and reconstruct the inputs from these representations [21]. By inputting a vector x into AE, a representation vector z in the latent layer (between encoder and decoder) can be compressed by the encoder neural network and reconstructed to the reconstructed vector $\hat{x}$ by the decoder neural network. The redundancies associated with unnecessary information can be eliminated during representation learning, while the most important information of the input data can be retained. Therefore, the reconstruction of the input vector x from the representation vector z is possible after training for several epochs. The proposal of VAE [22] rapidly promoted the development of AE technology to a great extent. The most significant difference between VAE and AE is that the representation vector z in the latent layer of VAE is a random variable with a Gaussian distribution, $N(\mu_{z},\sigma_{z}^{2})$. In addition, it is worth noting that while the reconstruction error typically serves as the primary loss function for AE, VAE introduces an additional loss component based on the Kullback–Leibler divergence (KLD) between the Gaussian distribution $N(\mu_{z},\sigma_{z}^{2})$ and the standard normal distribution N(0,I). This KLD loss term further enhances the capability to capture and model the latent space distribution.

### 3.4. Self-Organizing Map (SOM)

To address the challenges of SQA, SOM, a clustering method based on unsupervised learning, has been used in some papers [18,23]. SOM employs a two-layer neural network to map an *N*-dimensional input layer onto a two-dimensional output layer. The output layer consists of multiple neurons, each serving as a potential match for an input sample, thereby identifying a winning neuron. SOM training typically involves competitive learning. The QE of an input vector can be regarded as the Euclidean distance between that vector and its corresponding winning neuron. T. Kohonen et al. [14] noted that a smaller QE signifies a better alignment between the input sample and the trained SOM model [14]. By leveraging the QE of a set of input data, it becomes possible to identify anomalies within the data. For instance, it can be utilized to detect low-quality signal segments when the majority of signal segments exhibit high quality.

## 4. Proposed Method

As shown in Figure 2, four components make up the framework of the robust FHR estimation approach assisted by DUS SQA. These four blocks are described separately as follows.

### 4.1. Pre-Processing

In general, fetal heart activity occurs at frequencies below 500 Hz. In addition, frequencies below 25 Hz are primarily correlated with noise generated by fetal movement or equipment. Thus, we apply a bandpass filter to raw DUS signals with cutoff frequencies of 25 Hz and 500 Hz to focus on fetal heartbeats [7]. In addition, 2D spectrograms are generated using the short-time Fourier transform (STFT) with a window length of 64 ms and a step size of 1 ms. To generate integrated spectrum data from spectrograms, which are time series data, the spectra are integrated over the spectrograms in the frequency range [25, 500] Hz [24]. An example of filtered DUS data, spectrograms, and integrated spectrum data is shown in Figure 3.

### 4.2. FRRI Estimation

Figure 4 demonstrates the workflow for FRRI estimation from integrated spectrum data. To calculate the FRRI from the integrated spectrum data, the ACF of an integrated spectrum segment is calculated using a window of length *S*, which is set to 3.75 s in this work. The selection of a 3.75 s window was based on its standard usage in the computerized analysis of fetal non-stress tests [7]. This window length has been widely adopted and proven to yield consistent and reliable results in related studies. Before calculating the ACF, it is necessary to minimize the effects of noise spikes. To achieve this, we employ a spike removal algorithm, as introduced in [25], which effectively eliminates these undesirable noise spikes. In the calculated ACF, the maximum peak is detected from the range $\left[FRRI_{min},FRRI_{max}\right]$, which is assumed to be a reasonable range of FRRIs. To ensure that the peaks are not harmonic, an additional procedure is performed if there are more than two distinct peaks within a window. The value of $FRRI_{min}$ and $FRRI_{max}$ is set to be the same as the optimized values in [7]. We also use the algorithm proposed by [7] to identify whether a peak is harmonic and to determine the prominent peak. The time (the horizontal axis of the ACF) of the detected peak of the ACF can be considered as an approximate estimate $FRRI_{app}$ in the window of length *W*. To determine the FRRI for each heartbeat, we use the approximate estimate $FRRI_{app}$ to adjust the window width for the second calculation of the ACF. The adjusted window length $S_{new}$ is defined as shown in Equation (Equation 3), which includes two heartbeats.
(3)$$S_{new}=2\times FRRI_{app}+\Delta ,$$
where Δ in this case is set to $1/2\times FRRI_{app}$. Using the adjusted window, we apply the ACF again and use the same procedure as before to detect the peak in the ACF to derive the FRRI. Different from the peak detection of the first ACF, in the second ACF, the maximum peak is detected within the range $\left[FRRI_{app}-range,FRRI_{app}+range\right]$. The detected peak of the second ACF is the estimated FRRI in the new window, called $FRRI_{est}$. As the objective of the second ACF calculation and peak detection is to further refine the approximate estimate $FRRI_{app}$, it is expected that the value of $FRRI_{est}$ should be in close proximity to $FRRI_{app}$. After conducting several experiments, a value of 0.1 s is chosen as the value of range to ensure that $FRRI_{est}$ remains around $FRRI_{app}$. After estimating the $FRRI_{est}$ in this window, the window length *S* is initialized as 3.75 s for the next window. In addition, the window is shifted for half the time of the previous $FRRI_{est}$. Each subsequent window goes through the same operation to estimate FRRIs.

### 4.3. SQA

As shown in Figure 2, there are two procedures, including representation learning based on VAE and combined SQI generation based on SOM. In our work, representation learning from the integrated spectrum is on the basis of an FCN-based VAE neural network. The integrated spectrum data are slid through a 1.2 s window (average value of resized windows after the first ACF for FRRI estimation). Each time the window is slid, the window is moved to align the start time stamps of the resized window recorded during FRRI estimation. As shown in Figure 5, the input vector $x=[x_{1},x_{2},\ldots ,x_{L}]$ is an integrated spectrum segment that has been resampled to 1024 samples, while the output vector $\hat{x}=\left[{\hat{x}}_{1},{\hat{x}}_{2},\ldots ,{\hat{x}}_{L}\right]$ represents a reconstructed integrated spectrum segment. In addition, the VAE neural network is a slight modification of an AE neural network used for noise reduction in ECG signals [26], where there are several FCN layers in both the encoder and decoder parts. Every FCN layer comprises three key components: (1) convolutional layer, (2) batch normalization layer, and (3) activation layer based on the exponential linear unit (ELU) function. Together, these components form the building blocks of each FCN layer, enabling effective feature extraction and non-linear transformations within the network. Furthermore, two dense layers and a sampling function are used to link the encoder and decoder. The mean and variance of the latent vector z, called $\mu_{z}$ and $\sigma_{z}^{2}$, are generated by the dense layers. The sampling function is then used to generate z from $\mu_{z}$ and $\sigma_{z}^{2}$. In the encoder neural network, after five convolutional layers with a stride of 2, the input vector x is compressed to a vector of dimension 32 × 40. Using the dense layers and the sampling function, $\mu_{z}$, $\sigma_{z}^{2}$, z are all of dimension 32 × 1. The VAE decoder takes the latent representation vector z as input and generates the reconstructed vector $\hat{x}$ as output. We use transposed convolutional layers throughout the decoder neural network for upsampling. All convolutional and transposed convolutional layers have a kernel size 16, and all batch normalization layers have a batch size 64.

Since the purpose of VAE training is to minimize reconstruction errors and enable z to approximate standard normal distribution N(0,I) as much as possible, the VAE loss function is given as Equation (Equation 4):(4)$$Loss=\frac{1}{L}\sum_{l=1}^{L}{\left(x_{l}-{\hat{x}}_{l}\right)}^{2}+{KLD}_{\mu_{z},\sigma_{z}^{2}},$$
where *L* is the length of the input vector x and the reconstructed vector $\hat{x}$, ${KLD}_{\mu_{z},\sigma_{z}^{2}}$ is KLD between $N(\mu_{z},\sigma_{z}^{2})$ and N(0,I), which is given as follows:(5)$$\;{KLD}_{\mu_{z},\sigma_{z}^{2}}=\frac{1}{2}\sum_{i=1}^{d}\left(\mu_{z(i)}^{2}+\sigma_{z(i)}^{2}-\log\sigma_{z(i)}^{2}-1\right),$$
where the dimension of z is *d*. After 100 training epochs, the latent representation vector z captures significant information about fetal cardiac activities, facilitating precise reconstruction of the input data.

By the trained FCN-based VAE neural network, the integrated spectrum segments are processed to the representation vectors with the dimension of 32 × 1. $\mu_{z}$ is input to the SOM to compute the combined SQI (means combining a 32 × 1 vector to a single value), which is introduced as follows. Considering that high-quality DUS signal segments make up more than 80% of all segments in our dataset, it is important to note that there is a significant imbalance between these two types of low- and high-quality segments. This imbalance allows the identification of low-quality segments as a task for outlier detection. Within the trained SOM neural network, the outliers exhibit considerable distance from their corresponding winning neuron, in stark contrast to the majority of samples which are positioned in close proximity to their respective winning neurons. Therefore, for detecting outliers, we use QE as the output of the SOM neural network and calculate the combined SQI based on QE.

The combined SQI is defined based on the QE. A QE value can be computed for each integrated spectrum segment by feeding it into the trained SOM. Then, the QEs are normalized in the range [0, 1] based on the min–max normalization algorithm. The combined SQI based on the normalized QE is defined as follows:(6)$$SQI\left(x\right)=\left\{\begin{array}{cc} 1, & QE(\mu_{z}\left(x\right))<=th,\\ 1-\frac{QE(\mu_{z}\left(x\right))-th}{1-th}, & QE(\mu_{z}\left(x\right))>th, \end{array}\right.$$
where SQI(x) is the combined SQI generated from the input vector of VAE x, th is a threshold used to determine the quality of data that can be used to generate acceptably accurate FHR estimates. th is specifically set to the 9-th decile of all SQI values. This choice is based on the observation that approximately 90% of the segments within our dataset exhibit high quality. For different datasets, th can be set to different values. The combined SQI has a range of [0, 1], and the larger value corresponds to the higher quality.

### 4.4. FRRI Estimation Refinement

KF is a common and effective tool for estimating the optimal state of stochastic signals [27]. Inspired by previous studies [15,16], a KF is applied to the rough FRRI estimates generated from the FRRI estimation block explained in Section 4.2 to further rectify the FRRI estimation. Our model is defined as a first-order autoregressive process, where the time series estimates are determined by the linear regression of the preceding measurement, establishing a relationship between the current estimate and its previous measurement. In this process, there are two noise metrics: process noise $W\in R^{n}$ and measurement noise $V\in R^{n}$, where W∼N(0,Q) and V∼N(0,R). In addition, we set the coefficient state transition metrics to be unitary. To merge the combined SQI into the KF, we modify the measurement noise covariance metric $R_{m}$ using the non-linear weighting function [15] based on the combined SQI, as shown in Equation (Equation 7). The measurement $Z_{m}$ should be less trusted when the combined SQI is low.
(7)$$R_{m}=R_{0}\cdot e^{\frac{1}{SQI_{m}^{2}}-1},$$
where $SQI_{m}$ refers to the *m*-th combined SQI, while R0 represents the initial value of the measurement noise covariance. As SQIm approaches 0, indicating low quality, Rm tends towards infinity. This adjustment serves to decrease the Kalman gain Km, leading to reduced reliance on the current measurement compared to the previous measurement. Conversely, $R_{m}$ approaches $R_{0}$ while $SQI_{m}$ approaches 1, and $K_{m}$ is increased to reflect greater reliance on the current measurement. By incorporating this adaptive mechanism, the algorithm assigns less trust to measurements with lower quality, prioritizing the more reliable information for improved estimation accuracy.

We first use a conventional KF with fixed $R_{m}$ and $Q_{m}$ to smooth the rough FRRI estimates, where we calibrate $R_{0}=1$ and $Q_{0}=0.1$. In addition, a KF with changing $R_{m}$ based on Equation (Equation 7) is applied to refine the FRRIs with relatively large errors which are estimated from low-quality DUS segments. Through calibration, we assign $R_{0}=1$ and $Q_{0}=1$ for the second KF. In addition, we then eliminate the FRRIs from where the low-quality segments are continuous since it is difficult to correct FRRIs generated from such signals. To detect low-quality segments, a combined SQI threshold is established, specifically the 8th decile of all combined SQI values. Any FHR estimates obtained from more than three consecutive segments of low quality are discarded.

## 5. Results and Discussion

### 5.1. Experimental Setup

The DUS recordings and invasive fetal ECG recordings of ten subjects are collected simultaneously by Atom Medical Corporation. The fetal R-peaks are annotated by the experts to obtain the ground truth of the FRRIs and FHRs. The relationship between an FHR and an FRRI is given by
(8)$$FHR_{i}=\frac{60\cdot F_{s}}{FRRI_{i}}$$
where $F_{s}$ is the sampling frequency of the DUS signal recordings. For each subject, there is a DUS recording of 60 s with a sampling frequency of 1 kHz. As for VAE, we used the Adam optimizer and trained for 100 epochs. In addition, to achieve optimal performance of the SOM training, we experimented with different values of each parameter while keeping the other parameters constant. A 30 × 30 output space and the Gaussian neighborhood function are used to train the SOM neural network for 15,000 iterations. The number of training samples for VAE and SOM neural network is 2466, where each training sample is a 1.2 s integrated spectrum, as mentioned in Section 4.3.

To evaluate the performance of the proposed robust FHR estimation method, three performance metrics are considered, which are given below:Root mean square error (RMSE): RMSE is calculated between the estimated value and the ground truth value of FRRI, which is calculated as follows:
(9)$$RMSE=\sqrt{\frac{1}{\Gamma }\sum_{i=1}^{\Gamma }\left|\right|{\hat{FRRI}}_{i}-FRRI_{i}{\left|\right|}_{2}^{2}},$$Averaged absolute error (AAE): AAE is calculated between the estimated value and the ground truth value of FHR, which is calculated as follows:
(10)$$\begin{array}{cc} AAE & =\frac{1}{\Gamma }\sum_{i=1}^{\Gamma }\left|\right|{\hat{FHR}}_{i}-FHR_{i}\left|\right|\\ \; & =\frac{1}{\Gamma }\sum_{i=1}^{\Gamma }\left|\right|\frac{60\cdot F_{s}}{{\hat{FRRI}}_{i}}-\frac{60\cdot F_{s}}{FRRI_{i}}\left|\right|, \end{array}$$Coverage: Since some rough FRRI estimates may be removed based on the results of the SQA, the ratio of reserved FRRI estimates to all FRRI estimates is a critical indicator of whether as many FRRI estimates as possible have been retained.

We compared these performance metrics in the following five scenarios:Use the method proposed by Valderrama et al. [7].FHR estimation only: We performed only two steps to obtain rough FRRI estimates, including preprocessing and FRRI estimation (described in Section 4.1 and Section 4.2).Remove unreliable FRRIs: We eliminate the unreliable rough FRRIs if the corresponding combined SQI is less than the 8th decile of all combined SQI values.Use conventional KF: A conventional KF with a fixed noise covariance metric was applied to the rough FRRI estimates.Proposed method: The four steps described in Section 4 are all used to estimate and refine FRRIs.

### 5.2. Experimental Results

The changing pattern of training loss throughout the training process of the FCN-based VAE is illustrated in Figure 6. In this paper, our model operates as a reconstruction model employing the principles of a VAE. The experimental results reveal a compelling trajectory in the training loss throughout 100 epochs. From 1 to 100 epochs, the training loss consistently decreases, underscoring the proficiency of the model in learning and capturing intricate patterns within the dataset. As a reconstruction model based on VAE, this observed reduction in loss attests to the success of the model in encoding and subsequently reconstructing input data.

To demonstrate the performance of the combined SQI, inspired by [28], we defined a normalized absolute error (NAE) calculated as Equation (Equation 11):(11)$$NAE=\frac{|FRRI_{est}-FRRI_{true}|}{FRRI_{true}},$$
where $FRRI_{true}$ and $FRRI_{est}$ represent the estimated FRRI and the ground truth of FRRI. By establishing a threshold, we broadly categorize all integrated spectrum segments into high-quality and low-quality. In our experiments, by manually observing low-quality and high-quality segments using different thresholds, we selected the most appropriate threshold of 0.03. It is important to note that the classification into high and low quality is not based on expert annotations but rather roughly annotated based on the assumption that “if the signal quality is low, the FRRI prediction is inaccurate”. Figure 7 illustrates the combined SQI corresponding to high-quality and low-quality segments. Since the number of low-quality segments is significantly less than that of high-quality segments, we randomly selected an equivalent number of low-quality segments from high-quality signals for illustration. From the figures, it is evident that the majority of combined SQI values for high-quality segments are equal to or close to 1, while some low-quality segments exhibit considerably lower combined SQI values. Consequently, we can affirm that our proposed combined SQI is effective in distinguishing between high-quality and low-quality signals.

In Figure 8, we present an example of the pre-processed DUS signal segment, and the combined SQI generated from this segment (20–25 s of subject 9). The figure illustrates the presence of noise and relatively high amplitudes in both the filtered DUS signal and the integrated spectrum within the time range of 22.5–23.5 s. Consequently, the estimated FRRI based on such a signal segment is highly unreliable, leading to significantly lower composite SQIs compared to the pre- and post-periods. This observation strongly suggests that the combined SQI is closely associated with the quality level of the DUS signals. The figure highlights the importance of signal quality assessment in accurately estimating the fetal heart rate and emphasizes the role of the combined SQI as an indicator of DUS signal reliability.

Table 1 and Figure 9 provide insights into the RMSE of FRRI, AAE of FHR, and coverage in five scenarios. It is evident that without SQA, our FRRI estimation method outperforms the method proposed by Valderrama et al. [7] for the majority of subjects. However, there are a few exceptions (e.g., subjects 7 and 8) where our method may not yield better results. The other three scenarios considered in the analysis also contribute to the reduction of estimation errors to varying extents. In the “Remove unreliable FRRIs” scenario, the RMSE and AAE generally decrease, except for subjects 1, 4, and 10. However, it is important to note that this scenario leads to a significant reduction in coverage, especially for subject 9. In the “Use conventional KF” scenario, the FHR estimation errors are reduced while maintaining 100% coverage. This is achieved by applying a conventional KF that helps in smoothing out the rough FRRI estimates and handling sharp changes in the data. Comparing all the scenarios, our proposed robust FHR estimation method demonstrates the lowest RMSE and AAE. Additionally, our method maintains higher coverage for each subject compared to the conventional approach of removing unreliable FRRIs, which is commonly used for signal quality assessment tasks. Moreover, the experimental results validate the effectiveness of our proposed technique in enhancing FHR prediction across various scenarios. Notably, our method showcases significant improvements for recordings encompassing a substantial proportion of low-quality data (e.g., subject 9), as well as for recordings containing a smaller fraction of low-quality data (e.g., subject 1). These results highlight the robustness of our approach, showcasing its ability to deliver reliable FHR predictions across recordings with varying levels of data quality.

### 5.3. Limitation and Future Works

Several limitations still exist, although our proposed FHR estimation approach can accurately estimate FHR from DUS signals and maintain robustness under different signal qualities. To train VAE and SOM neural networks for the DUS SQA part, we use all signal segments for training without a selection procedure. In fact, it is worth mentioning that this strategy only works if the signal segments are mostly of high quality, in other words, low-quality segments can be considered as outliers. If low-quality signal segments account for more than 40% of a dataset, it is better to use some simple quality indicators to roughly select high-quality data for training (e.g., entropy, skewness, or kurtosis). In addition, we specifically developed and tested our methods using only one dataset consisting of a small number of subjects, optimizing parameters for this dataset without retaining data for validation, so it is not possible to verify that the developed algorithms would be generalizable to the larger population of subjects studied. For our future work, some larger datasets are needed to further verify the robustness.

## 6. Conclusions

We present a robust DUS-based FHR estimation method assisted by DUS SQA based on unsupervised representation learning. By using STFT to generate integrated spectrum data, information on DUS signals in both time and frequency domains is extracted simultaneously. To achieve accurate FRRI estimation, we employ two ACF calculations on the integrated spectrum data. The first ACF is utilized for approximate FRRI estimation and window resizing, while the second ACF provides a more precise FRRI estimation. The length of the FRRI estimation window is adjusted based on the previously estimated FRRI, enabling adaptability to varying heart rates. In addition, we propose a DUS SQA method that uses an FCN-based VAE to obtain learning representations and a SOM neural network for combined SQI generation. The combined SQI is used to modify the measurement noise covariance, and a KF is then used to refine the FRRIs estimated from low-quality signals. Through extensive experiments, we demonstrate the superiority of our proposed approach in terms of accuracy and robustness for DUS-based FHR estimation, surpassing conventional methods. Moreover, our SQA approach can be extended to address various SQA challenges involving different types of signals, as it is built entirely on unsupervised learning techniques.

## Figures and Tables

**Figure 1 sensors-23-09698-f001:**
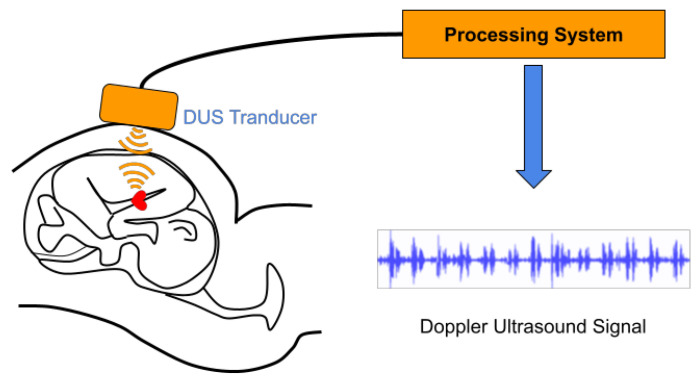
DUS signal measurement. The US transducer is placed on the maternal abdomen to extract DUS signals.

**Figure 2 sensors-23-09698-f002:**
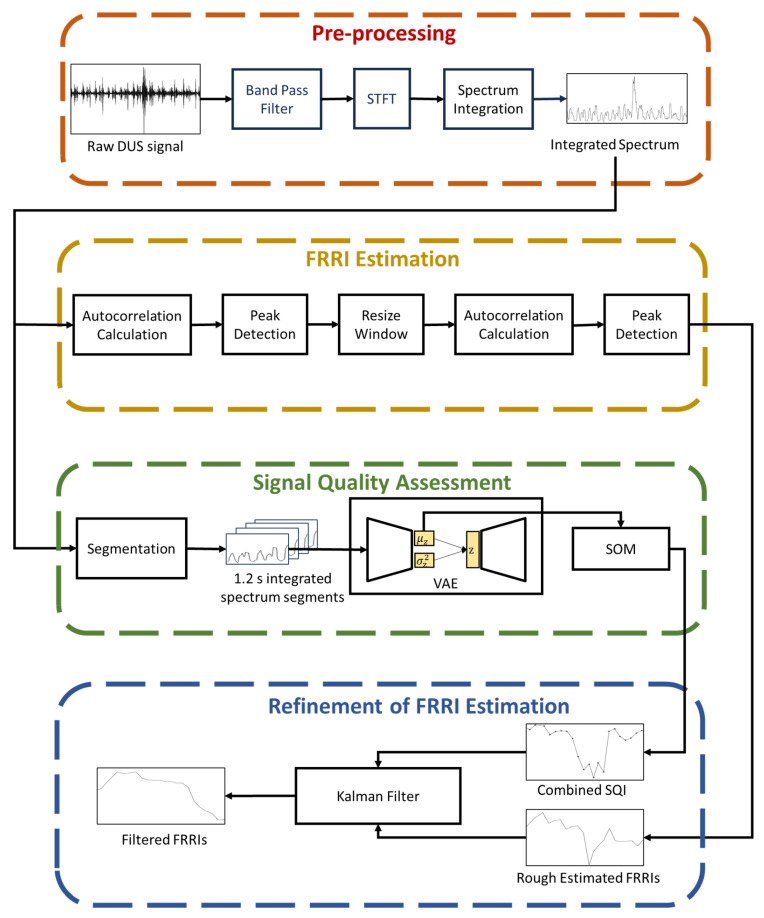
The overview of the proposed method. Four components make up the framework of the robust FHR estimation approach assisted by DUS SQA: pre-processing, FRRI estimation, SQA, and refinement of FRRI.

**Figure 3 sensors-23-09698-f003:**
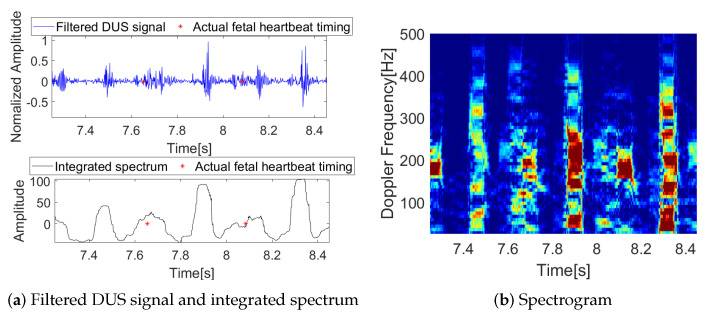
An example of filtered DUS data, spectrograms, and corresponding integrated spectrum data. The actual fetal heartbeat timings (R-peaks) are marked in red.

**Figure 4 sensors-23-09698-f004:**
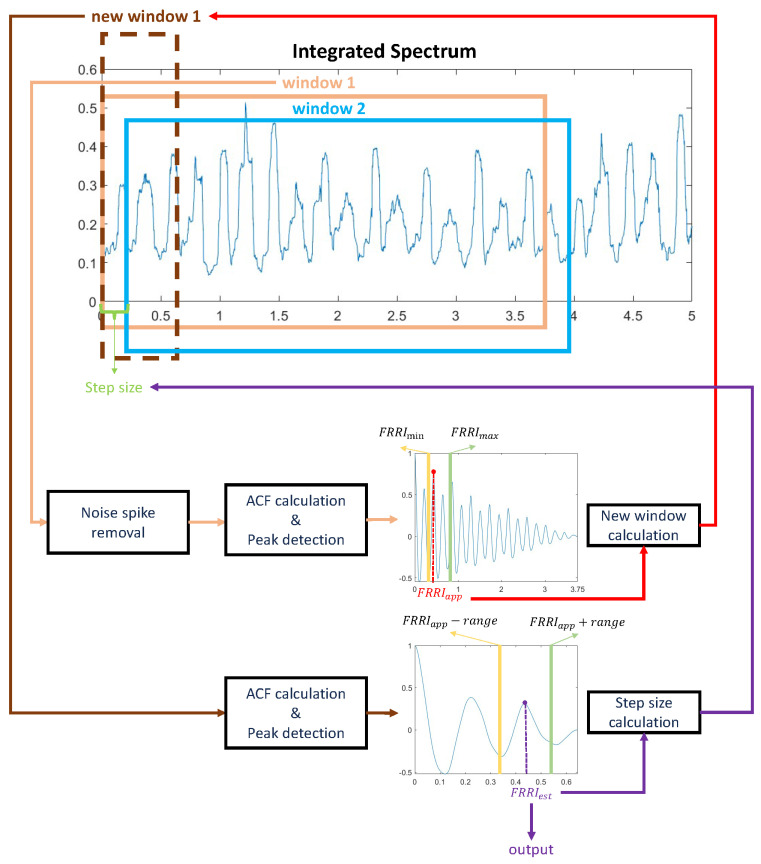
The workflow for FRRI estimation. This figure only presents the workflow for estimating FRRI in window 1 and determining window 2. The FRRI estimation procedure for each following window follows this workflow.

**Figure 5 sensors-23-09698-f005:**
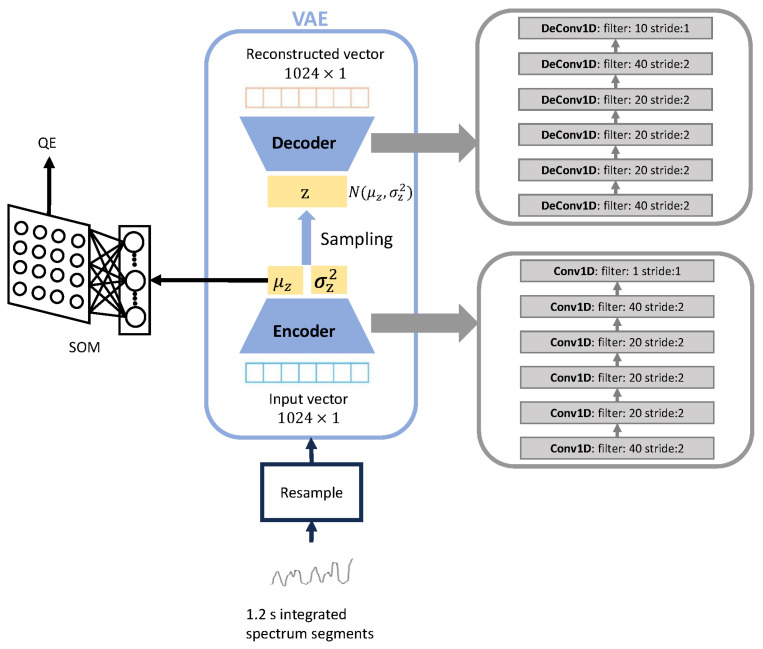
The proposed VAE and SOM-based SQA. Both the encoder and decoder of VAE have multiple FCN layers. Each FCN layer consists of a convolutional layer, a batch normalization layer, and an exponential linear unit-based activation layer. Two dense layers and a sampling function are used to connect the encoder and decoder.

**Figure 6 sensors-23-09698-f006:**
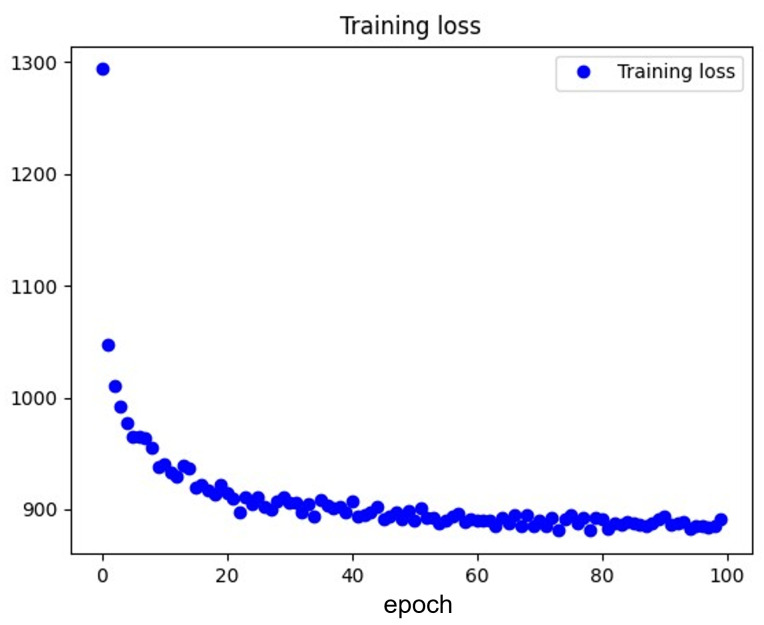
The training loss of the FCN-based VAE training process. The VAE neural network is introduced in Section 4.3.

**Figure 7 sensors-23-09698-f007:**
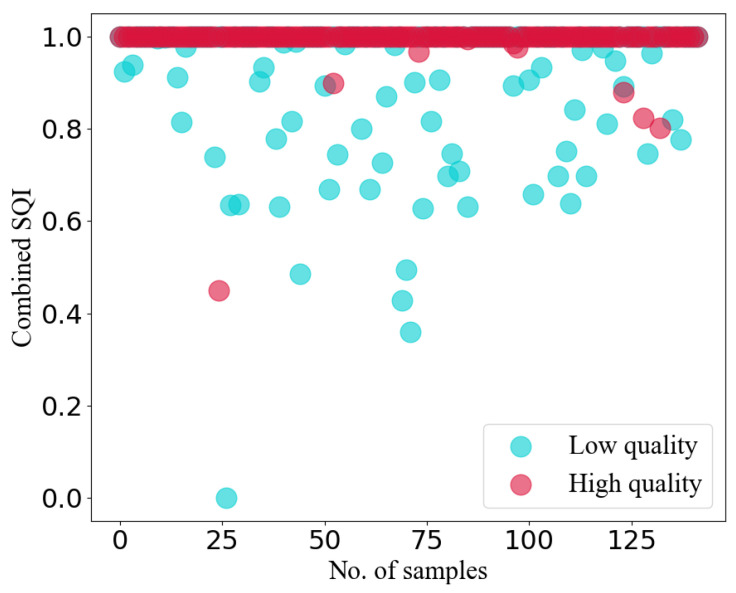
The combined SQI values of the low- and high-quality integrated spectrum segments.

**Figure 8 sensors-23-09698-f008:**
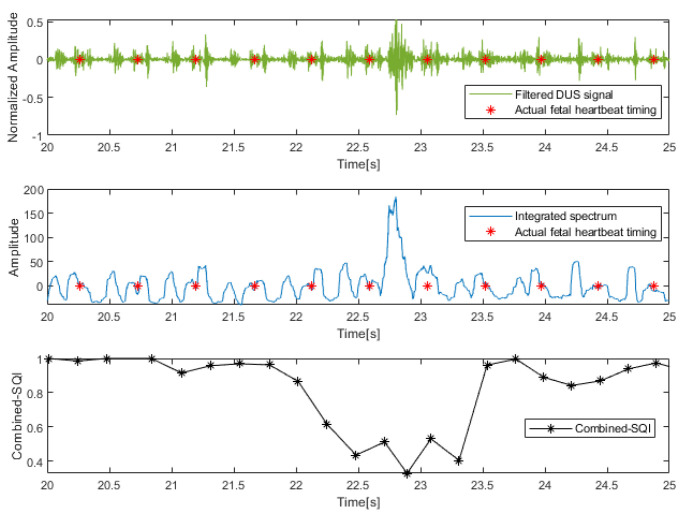
The filtered DUS signal, the integrated spectrum, and the combined SQI values (from top to bottom) of a 5 s signal segment (20 s–25 s of subject 9).

**Figure 9 sensors-23-09698-f009:**
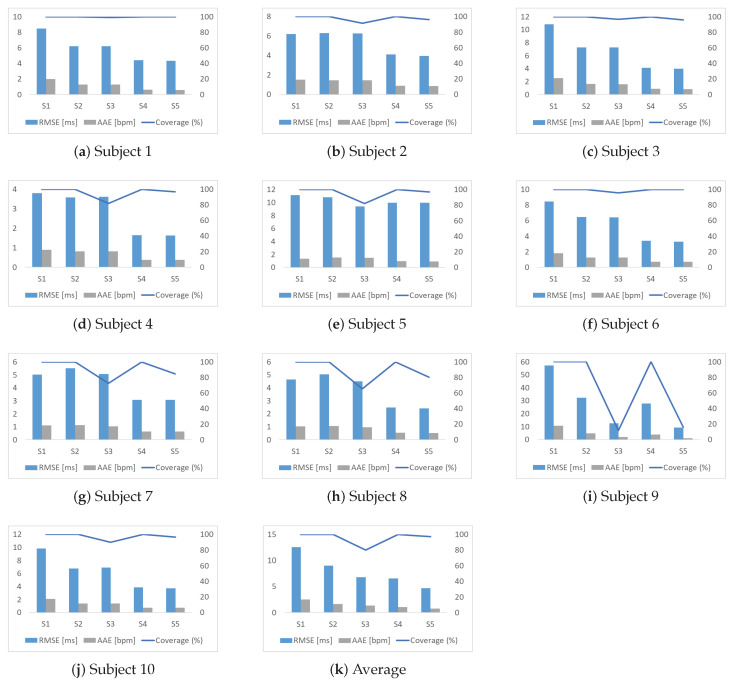
Comparison of performances of five scenarios for each subject. S1: Use the method proposed by Valderrama et al. [7]; S2: FHR estimation only; S3: Remove unreliable FRRIs; S4: Use conventional KF; S5: Proposed method. RMSE of FRRI, AAE of FHR, and coverage for each subject and average values are shown in this figure. In each sub-figure, the y-axis on the left corresponds to the values of RMSE and MAE, and the one on the right corresponds to the values of coverage.

**Table 1 sensors-23-09698-t001:** Comparison of performances of five scenarios: (1) Use the method proposed by Valderrama et al. [7]; (2) FHR estimation only; (3) Remove unreliable FRRIs; (4) Use conventional KF; (5) Proposed method. RMSE of FRRI, AAE of FHR, and coverage for each subject and average values are shown in this table.

		1	2	3	4	5	6	7	8	9	10	Average
Valderrama et al. [7]	RMSE [ms]	8.46	6.21	10.81	3.80	11.12	8.46	5.02	4.64	57.10	9.83	12.54
	AAE [bpm]	1.98	1.51	2.54	0.90	1.34	1.80	1.11	1.04	10.79	2.09	2.51
	Coverage (%)	100.00	100.00	100.00	100.00	100.00	100.00	100.00	100.00	100.00	100.00	100.00
FHR estimation only	RMSE [ms]	6.20	6.30	7.26	3.58	10.83	6.48	5.52	5.03	32.36	6.78	9.03
	AAE [bpm]	1.27	1.47	1.63	0.82	1.55	1.27	1.14	1.06	4.89	1.39	1.65
	Coverage (%)	100.00	100.00	100.00	100.00	100.00	100.00	100.00	100.00	100.00	100.00	100.00
Remove unreliable FRRIs	RMSE [ms]	6.22	6.27	7.25	3.62	9.42	6.43	5.06	4.50	12.62	6.90	6.83
	AAE [bpm]	1.27	1.45	1.61	0.82	1.48	1.26	1.04	0.97	2.06	1.40	1.34
	Coverage (%)	99.23	91.41	96.87	81.93	95.18	95.53	72.54	65.52	11.95	89.75	79.99
Use conventional KF	RMSE [ms]	4.42	4.12	5.15	1.64	9.96	3.40	3.07	2.49	27.77	3.87	6.59
	AAE [bpm]	0.63	0.90	1.01	0.37	0.96	0.73	0.64	0.54	3.83	0.74	1.03
	Coverage (%)	100.00	100.00	100.00	100.00	100.00	100.00	100.00	100.00	100.00	100.00	100.00
Proposed method	RMSE [ms]	4.35	3.97	5.00	1.62	9.95	3.31	3.07	2.41	9.22	3.75	4.67
	AAE [bpm]	0.59	0.86	0.98	0.37	0.94	0.72	0.63	0.52	1.07	0.72	0.74
	Coverage (%)	100.00	96.06	100.00	96.77	100.00	100.00	84.71	80.35	16.00	96.69	87.06

## Data Availability

The data presented in this study are from Atom Medical Corporation. The data are not publicly available.

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
