# Peer review of "A Robust Approach Assisted by Signal Quality Assessment for Fetal Heart Rate Estimation from Doppler Ultrasound Signal"

_sensors, 2023, doi:10.3390/s23249698_

Round 1

Reviewer 1 Report

Comments and Suggestions for Authors

The paper is well-written and the methodology appears to be sound and to bring some improvement with respect to other existing methods.

Comparing the results in table 1 proves quite cumbersome. I would suggest to find a way to illustrate the results (also) in a graphical manner.

The main limitation of the study, as correctly recognized by the authors, consists in the limited number of subjects used in the tests and in the absence of a statistical validation phase.

Minor:

p2: "Also, sixty divided by an FRRI equals an FHR" sounds a bit trivial, perhaps it is better to remove or reformulate this sentence.

p2: It is important to discuss in detail the differences adn extensions of the present work with respect to the conference paper [17].

p11, l329: the number of the table is missing

Reviewer 2 Report

Comments and Suggestions for Authors

Authors used DUS to detect FHR based SQI assessment. The contribution of this study is the proposed VAE, a deep learning model to estimate SOI of each FHR. Authors clearly described their method. But, there are the litter problems in the texts.

1.     DUSs are the open dataset. Authors should describe this dataset and its web.

2.     In FC-ACE, authors should describe the number of training and testing samples.

3.     VAE is a supervise learning, how to label the target output should be mentioned.

4.     The training process curves should be mentioned.

5.     In Eq. (6), SOI is defined as a value. Its performance should be mentioned. Please refer “Classification of photoplethysmographic signal quality with deep convolution neural networks for accurate Measurement of cardiac stroke volume”.

6.     In sector 3, preliminaries should be uppercase “p”.

7.     In Fig. 2, the caption of second block does not fit the text mention.

8.     In the second paragraph of 5.2 section, the number of table is losing.  

Comments on the Quality of English Language

Authors should revise the spelling of texts again. 

Reviewer 3 Report

Comments and Suggestions for Authors

This articles entitledA Robust Approach Assisted by Signal Quality Assessment for Fetal Heart Rate Estimation from Doppler Ultrasound Signal“ presented  conclusion of investigation; . „By incorporating the SQI and Kalman filter (KF), we 13 refine the estimated FHRs, minimizing errors in the estimation process. Experimental results demonstrate that 14 our proposed approach outperforms conventional methods in terms of accuracy and robustness“.

The presenters presented the given objectives of the work with excellent graphic, pictorial and numerical representations, and comments in the components of the article, along with a discussion of similar research and citations of adequate fresh literature.

1. What is the main question addressed by the research?

Described in Title and Methodology

2. Do you consider the topic original or relevant in the field? Does it address a specific gap in the field?

Yes

3. What does it add to the subject area compared with other published material?

Original new results

4. What specific improvements should the authors consider regarding the methodology? What further controls should be considered?

No need

5. Are the conclusions consistent with the evidence and arguments presented and do they address the main question posed?

Yes

6. Are the references appropriate?

Yes, but please please edit according to the rules of the journal !!

7. Please include any additional comments on the tables and figures.

Without comments

Reviewer 4 Report

Comments and Suggestions for Authors

The contribution of this paper needs to explain clearly in the abstract. The abstract must highlight the key findings of the paper

Author has to highlight the scope of the research work and mentioned the importance of their research

Author has to highlight the research gap in literature survey, how the previous works carried out and performance of the existing methods

How the parametric analysis is carried out for various features to show the enhancement in the proposed algorithm

The topic studied in this paper has been extensively investigated. The advantages of the current work compared with existing ones should be further emphasized. Also Furnish discussion to establish how the proposed method is unique or different from other existing methods

Experimental discussion section should emphasize the used performance metrics further

Round 2

Reviewer 2 Report

Comments and Suggestions for Authors

No comment!